# Influence of Block Ratio on Thermal, Optical, and Photovoltaic Properties of Poly(3-hexylthiophene)-*b*-poly(3-butylthiophene)-*b*-poly(3-octylthiophene)

**DOI:** 10.3390/molecules27238469

**Published:** 2022-12-02

**Authors:** Van Hai Nguyen, Thanh Danh Nguyen, Jongwoo Song, Jongdeok An, Chan Im

**Affiliations:** 1Department of Chemistry, Konkuk University, 120 Neungdong-ro, Gwangjin-gu, Seoul 05029, Republic of Korea; 2Institute of Chemical Technology, Vietnam Academy of Science and Technology, 1A, TL29, Thanh Loc Ward, District 12, Ho Chi Minh City 700000, Vietnam

**Keywords:** triblock copolymer, GRIM polymerization, poly(3-alkylthiophene), photoluminescence quantum yield, photovoltaic power conversion efficiency

## Abstract

Efforts to improve the solar power conversion efficiencies of binary bulk heterojunction-type organic photovoltaic devices using an active layer consisting of a poly-(3-alkylthiophene) (P3AT) homopolymer and a suitable fullerene derivative face barriers caused by the intrinsic properties of homopolymers. To overcome such barriers, researchers might be able to chemically tailor homopolymers by means of monomer ratio-balanced block copolymerization to obtain preferable properties. Triblock copolymers consisting of three components—3-hexylthiophene (HT), 3-butylthiophene (BT), and 3-octylthiophene (OT)—were synthesized via Grignard metathesis (GRIM) polymerization. The component ratios of the synthesized block copolymers were virtually the same as the feeding ratios of the monomers, a fact which was verified using ^1^H-NMR spectra. All the copolymers exhibited comparable crystalline and melting temperatures, which increased when one type of monomer became dominant. In addition, their power conversion efficiencies and photoluminescence properties were governed by the major components of the copolymers. Interestingly, the HT component-dominated block copolymer indicated the highest power conversion efficiency, comparable to that of its homopolymer, although its molecular weight was significantly shorter.

## 1. Introduction

Regioregular poly(3-alkylthiophenes) (rr-P3ATs) with narrower bandgaps and higher electrical conductivities have been reported by various research groups [1,2,3]. P3ATs were synthesized by Kumada cross-coupling with a Ni(dppp)Cl_2_ catalyst via a chain-growth mechanism [4,5] and Grignard metathesis (GRIM) [6,7,8]. Among the P3ATs, the rr homopolymer bearing a hexyl side chain, known as P3HT, exhibited the highest power conversion efficiency (PCE). The PCE value of approximately 5% was significantly higher than those of poly(3-butylthiophene) (P3BT) and poly(3-octylthiophene) (P3OT) [9]. This was attributed to the higher charge carrier extraction capability of the P3HT active layer which is blended with a proper fullerene derivative, owing to its preferable molecular morphology after undergoing the annealing process [10,11]. P3BT might be a suitable material for organic photovoltaic (OPV) applications owing to its higher crystallization facility [12]. However, most studies have demonstrated unsatisfactory results for P3BT-based devices owing to their poor solubility [11]. Therefore, efforts have been made to increase their solubility by achieving the beneficial properties of preferable stacking.

One possible synthetic method to achieve higher solubility is the incorporation of longer alkyl side-chain thiophene moieties into P3BT to form block copolymers. Thus, P3HT and P3OT were selected because of their better solubility compared to the P3BT homopolymer and the possibly higher charge carrier mobility, as reported previously [10]. However, random copolymerization must hinder the achievement of a sufficiently high regioregularity of P3ATs, which must cause unfavorable disorder effects by forming active layers of devices with an optimized nanomorphology. Block copolymers with high regioregularity synthesized via quasi-living chain polymerization have attracted significant attention in the investigation of OPV devices [13,14]. An rr copolymer prepared from thiophene and selenophene derivatives showed phase separation and domain segregation, which could be pronounced with these block copolymers [15]. A diblock copolymer of poly (3-butylthiophene)-b-poly (3-octylthiophene) (P3BT-*b*-P3OT) was synthesized via the GRIM method, and its phase behavior was investigated, which confirmed the microphase separation of diblock copolymers, resulting in two distinct crystalline domains with lamellar structures [16]. Diblock copoly(3-AT)s, based on 3-butylthiophene, 3-hexylthiophene, and 3-dodecylthiophene, were synthesized via the GRIM method, and their microphase morphologies were investigated. The characterization of P3BT-*b*-P3HT revealed its ability to co-crystallize to form a uniform crystal domain [17]. P3HT-*b*-P3OT exhibited a good photovoltaic PCE and high tensile elasticity [12]. This is plausible because the long side chains of P3ATs can enhance the diffusion of [6,6]-phenyl-C_61_-butyric acid (PCBM) into the polymer matrix, which can possibly disturb the formation of optimum phase separation and subsequently decrease the PCEs of OPV devices [18].

Moreover, the synthetic strategy of using block copolymerization continues to expand, not only with diblock copolymers, but also with triblock copolymers for various organic electronics. Indeed, the triblock copolymer poly(3-dodecylthiophene)-*b*-polythiophene-*b*-poly(3-dodecylthiophene) (P3DDT-b-PT-b-P3DDT) was synthesized by the addition of monomers into the sequence via the GRIM method [8]. Miyane et al. [19] prepared ABA triblock copolymers based on poly(1,1,1,3,3,5,5-heptametyl-5-(6′-(thien-3″-yl)hexyl) trisiloxane-2″,5″-diyl) (P3SiHT) and P3HT. The formation of phase-separated P3HT and P3SiHT domains in the triblock copolymer (P3HT-*b*-P3SiHT-*b*-P3HT) was observed. Recently, the structures of triblock copoly(3-alkylthiophene)s, including P3BT, P3HT, P3OT, and P3DDT with proper block sequences, have been investigated. The results of these tests indicated stable charge mobilities at high temperatures and good flexibility in organic field-effect transistor (OFET) devices [20]. 

However, accurate studies on the feasibility of these triblock copolymers as active OPV materials, in conjunction with their underlying operating mechanisms, have not been performed. Therefore, in the present work, structurally well-defined rr-P3AT triblock copolymers with different block ratios were synthesized using a modified GRIM method. To achieve effective solubility enhancement of P3BT, 3-butylthiophene (BT) blocks were spanned by the blocks of 3-hexylthiophene (HT) and 3-octylthiophene (OT) at the ends of both sides, that is, P3HT-*b*-P3BT-*b*-P3OT. These triblock copolymers were then used to investigate their quantitative thermal properties using conventional differential scanning calorimetry (DSC), followed by characterization of their fundamental optical properties, for example, UV-Vis absorption and photoluminescence spectra, in conjunction with their quantitative PL quantum yield (PLQY) estimation. These results provide insights into the effect of changing the block ratio on exciton behavior. Finally, the detailed data were compared with their OPV device parameters, including incident photon-to-current efficiency (IPCE) spectra, which were measured as bulk heterojunction-type devices by blending them with a proper fullerene derivative known as PCBM.

## 2. Results

### 2.1. Synthesis

The synthesis of the triblock copolymers of P3HT-*b*-P3BT-*b*-P3OT with various ratios of components was carried out via a modified GRIM method, as illustrated in Figure 1. P3HT block (2) was first synthesized by the polymerization of (1), followed by the addition of (3) to obtain (4), and finally, (5) was added to afford triblock copolymers [21]. Copolymers with different compositions (P1, B2, B3, H2, H3, O2, and O3) were fabricated by changing the feeding ratios of monomers containing hexyl, butyl, and octyl groups, as shown in Table 1. Before the effect of the block ratios on the physical and photovoltaic properties of the tri-block copolymers can be investigated, the block ratios in the synthesized copolymers must be accurately identified.

In the present work, the monomer conversion of 2-bromo-5-iodo-3-alkylthiophenes into 2-bromo-3-alkylthiophenes with Grignard reagents was determined before the initialization of the polymerization process. For the experimental procedure, all Grignard products (1, 3, and 5) were withdrawn by quenching with water after stirring for 2 h at 0 °C, and the organic layer extracted with diethyl ether was used for ^1^H-NMR spectroscopy (Appendix A). As shown in Appendix A, proton H_a_ peaks originating from 2-bromo-5-iodo-3-alkylthiophene (6.95 ppm) are not observed in any of the quenched products, confirming the high conversion performance of the monomer during the Grignard exchange reaction. 

To evaluate the ratio of the components in the synthesized copolymers, four different copolymers (P1, B3, H3, and O3) were analyzed by ^1^H-NMR, as shown in Figure 1. Compared with the spectra of the homopolymers shown in Appendix A, the triplet peaks centered at approximately 0.98 ppm could be assigned to the resonance of the terminal methyl group in butyl sidechain, and the multiple peaks centered at approximately 0.90 ppm were assigned to the terminal methyl groups in hexyl and octyl sidechains. However, the peaks of the terminal methyl group in the P3HT and P3OT blocks overlapped; therefore, it was difficult to determine the separate ratio of each block when the estimation was performed with only ^1^H-NMR spectral analysis. Nevertheless, the relative block size ratio between P3BT and the sum of P3HT + P3OT could be quantitatively estimated by integrating the peak areas centered at 0.98 and 0.90 ppm separately, as seen in Figure 1. The P3BT: (P3HT + P3OT) ratios of copolymers P1, B3, H3, and O3 were 1:1.98, 3:2.05, 1:3.98, and 1:4.01, respectively. These ratios were virtually the same as their feeding molar ratios of 1:2 (1:(1 + 1)), 3:2 (3:(1 + 1)), 1:4 (1:(3 + 1)), and 1:4 (1:(1 + 3)), respectively. Nevertheless, if the conversion yields of the initial monomer into block copolymers after the polymerization process are sufficiently high, then the ratio between P3HT and P3OT can be safely assumed from the mutual feeding ratio.

Therefore, the completeness of polymerization, that is, the degree of monomer conversion, was investigated using quantitative NMR (qNMR) analysis based on ‘internal standard’ referencing, as reported previously [22,23]. To apply this method, polymerization was performed with naphthalene as the internal standard. For example, three triblock copolymers, B3, H3, and O3, were investigated using qNMR. The peaks at 6.82 ppm and 7.11 ppm, being assigned to H_a_ and H_b_ of unreacted monomers, were integrated. They were then compared with those at 7.40 ppm and 7.77 ppm, being assigned to H_1_ and H_2_ of naphthalene, as shown in Appendix A. The results indicated that the polymerization degrees of B3, H3, and O3 after 24 h at room temperature were as high as 90%, 91%, and 93%, respectively. All parameters used to calculate the yields are listed in Appendix A. The polymerization degree increased with increasing alkyl side-chain length, which can be attributed to the better solubility of the copolymers containing longer alkyl chains in the reaction solvent of THF [10]. With the high yield of monomer conversion, it can be confirmed that the actual ratio of each P3AT block in the triblock copolymers was virtually the same as the feeding molar ratio of the starting monomers. 

Moreover, ^1^H-NMR data showed that regioregularity of the copolymer was estimated to be approximately 98% using the NMR spectra (Appendix A). GPC was employed to determine the copolymer molecular weight (MW) and polydispersity index (PI), and the results are shown in Table 1. Notably, the values of Mn and Mw increased when the monomer type became dominant in the triblock copolymer, as in the case of H3, B3, and O3. This may be related to the increasing trend of the polymerization degree with the extension of the side-chain length. It is plausible that Mw increases upon increasing the polymerization degree. P3BT polymerization suffers from poor solubility, as reported previously [10,24].

### 2.2. Thermal Analysis 

The thermal properties of all copolymers were investigated using DSC curves. The samples were heated to 350 °C and cooled to 30 °C at a scanning rate of 10 °C/min. The endo- and exothermic traces of the second scan are shown in Figure 2, and the extracted crystallization temperatures (T_c_) and melting temperatures (T_m_) are listed in Table 2. The thermal peak patterns of all the copolymers showed a single peak, independent of the block ratio of the copolymers. These results confirmed that the block copolymers underwent co-crystallization, something which was in agreement with a previous report on block copolymers of P3BT-*b*-P3HT [16]. The T_c_ and T_m_ values of H3, B3, and O3 with a dominant block component are comparable to those of their corresponding homopolymers, as reported in our previous study [10]. 

An increase in the component mixing degree of the copolymers (from ‘B3, H3, and O3’ to ‘B2, H2, and O2’) induced a significant reduction in T_c_ and T_m_, although this feature was not observed in the case of OT-dominant triblock copolymers. Nevertheless, T_c_ and T_m_ were further reduced upon increasing the mixing degree of the block components in the copolymers compared to P1. The reason why the OT block copolymers did not reduce their T_c_ and T_m_ can be explained by the fact that P3OT possesses the lowest T_c_ and T_m_ values among those of the P3ATs; therefore, the reduction was limited to those of the OT-dominated copolymers. This provides an insight into the thermal behavior of copolymers in which copolymerization does not appear to form any additional phase, causing a drastic change in their thermal characteristics, although the copolymerization sensitively affects their co-crystallization trends spanned by the T_c_ and T_m_ ranges of the incorporating blocks.

### 2.3. Optical Properties

#### 2.3.1. Pristine Films of Copolymers

The optical properties of all copolymers were investigated using UV-Vis and PL spectroscopy. All copolymers in CB solutions showed similar maximum absorption peaks at approximately 453 nm and maximum PL emission peaks at approximately 580 nm owing to the same optically active thiophene moieties and the absence of strong interactions between polymer chains in the solution, as shown in Figure 3A. The UV-Vis and PL spectra of the pristine films are shown in Figure 3B. The UV-Vis and PL spectra of the pristine films were significantly red-shifted compared to the mutual maximum peak positions of the corresponding solutions. The fine structures were more pronounced in the spectra of the pristine film samples than in the solution spectra. Two peaks centered at ~522 nm and ~550 nm, and one shoulder peak around ~600 nm, were observed. Obviously, H3 dominated copolymer had the most pronounced fine structure intensities. These spectroscopic results for the block copolymers are analogous to their corresponding homopolymers [10], particularly when one type of monomer is dominant. 

#### 2.3.2. Blend Films of Copolymers

The optical absorption spectra of the blended films were characterized by UV-Vis spectroscopy before and after thermal annealing at 140 °C for 15 min, as shown in Figure 4. The non-annealed thin films of these block copolymers exhibited broad absorption bands between ~400 and ~650 nm. An additional absorption band of PCBM is observed in the spectra below ~600 nm [25]. The absorption spectra of the blended films before annealing were blue-shifted and had less pronounced fine structures compared to those of the pristine thin films. This can be related to the relatively homogeneous dispersion of PCBMs between the polymer chains, which hinders effective interchain interactions [26]. An enhancement in the fine structure was observed after annealing, as shown in Figure 4. The pronounced shoulder peaks revealed that the higher ordering of the polymer chains resulted in enhanced preferable interactions between the polymer chains [27,28]. Undoubtedly, the process of thermal annealing triggers the formation of an optimum nanomorphology, which can be seen by the pronounced fine structures in their UV-Vis absorption spectra. Furthermore, this feature, which induced by annealing, is closely correlated with PL quenching and PCE improvement in the binary BHJ-type active layer in OPV devices. The effects of the absorption spectra on the annealing of blended copolymers were similar to those of homopolymers, although the changes upon annealing of copolymers were moderate because the triblock copolymers have averaged the characteristics of the three species, BT, HT, and OT.

The PL spectra of the pristine and blended films under ambient conditions were obtained by excitation at the maximum wavelengths of the absorption bands, as shown in Figure 5. The PL intensity of H3 block copolymers reduces by 87% upon PCBM addition, being slightly more than the reduction in P1 (80%). The reduction in the PL intensity upon PCBM blending of copolymer films reveals that the primary excitons generated in electron-donating copolymers were effectively quenched owing to the well-dispersed electron-accepting PCBM, which formed an intercalated phase with the copolymers [29,30]. Subsequent annealing of the blend films further reduced the residual PL intensities, which implies that the exciton dissociation facility of the blend films is enhanced by the nanomorphology owing to microscopic phase separation and higher ordering of copolymer backbones upon thermal annealing. This PL feature upon annealing is virtually the same as that of the corresponding homopolymers, as reported previously. [10] It is noteworthy that the PL intensity of the P1 blend film was lower than that of other block copolymers, such as B3, H3, and O3, even before annealing. This is an interesting block ratio-dependent PL quenching behavior, although the PL intensities of all the measured copolymers were similar after annealing. 

Quantitative analysis of the PL intensity is crucial, particularly for PV applications. This is because PL, the radiative recombination of primary excitons, is one of the most important decay paths that compete with the dissociation of primary excitons, at least at the initial state of free charge carrier formation. Therefore, the accurate PL quantum yield (PLQY) can be a measure of exciton dissociation, possibly followed by exciton diffusion to a donor/acceptor interface [29,31]. Therefore, the block copolymers were investigated using an integrating sphere equipped with a PLQY measurement, which is described in detail elsewhere [32]. As shown in Figure 6, all PLQYs of the dilute solutions were virtually the same, with a value of ~0.3, as mentioned previously. However, these values were approximately 10% lower than those of the homopolymers (~0.33), as reported previously [10]. The PLQYs of the pristine films both before and after annealing were decreased by a greater order of magnitude than those of the corresponding diluted solutions, where intense interchain interactions cannot be expected. The PLQYs increased when one monomer unit became dominant compared to the other units, such as B3, H3, and O3. The highest PLQY values (~0.03) are comparable to those of the homopolymers before annealing. However, the copolymer with a 1:1:1 block ratio, P1, exhibited the smallest PLQY of ~0.01. This might be a sign that block copolymers have a type of exciton dissociation site which correlates with the heterojunction between the different block types that can be activated in the solid state but not in the diluted solution. Furthermore, this impressive block ratio-dependent PLQY behavior of pristine films becomes weaker after the annealing of those pristine films. The PLQY of the copolymer films was further reduced to ~0.004 upon blending with the PCBM acceptor, virtually independent of the block ratio of the copolymers, and the unusual block ratio dependence of the unannealed pristine films became much weaker, as shown in Figure 6. 

### 2.4. Device Characteristics

The J–V characteristics of all the block copolymer devices are shown in Figure 7, and the parameters of the annealed devices are summarized in Table 3. The V_OC_ of all devices decreased slightly; however, the J_SC_ and fill factors (FF) significantly increased after thermal annealing because of the steeper slope of the J–V curves upon annealing. The devices with BT-rich (B2 and B3) blend films have lower J_SC_ values than those of HT, while those of OT have higher values than HT before annealing. Interestingly, BT-rich films showed a drastic increase in J_SC_ upon annealing, whereas those of HT increased moderately and OT only increased marginally. This increase in J_SC_ upon annealing contributes to the improvement in the PCE values because the PCE is related to the FF and J_SC_ with the following equation, PCE = (FF × J_SC_) / (total incident power). In addition, the increase in the FF upon annealing also contributed significantly to the PCE improvement in conjunction with the analog functional dependence between the FF and PCE. The PCE improvement based on the increase in both J_SC_ and FF reveals that annealing not only improved the charge carrier generation efficiency but also the charge carrier mobility as well as the extraction facility, as discussed previously [33]. Most features of the PCE improvement upon annealing of copolymer-based devices are similar to those of homopolymers. The PCE improvement in the P3HT homopolymer was also contributed to by FF and J_SC_. However, the MW-sensitive P3HT homopolymer could reach its highest PCE of 3.6% with MW of 39 kDa, as reported in our previous study [10], whereas the H3 copolymer with MW of 15 kDa has shown a PCE of 4.1%. Nevertheless, the FFs and the overall PCEs of copolymer:PCBM BHJ devices increased with increasing block ratio, from H2 to H3, B2 to B3, and O2 to O3 as shown in Figure 8.

The IPCE spectra of the devices, before and after annealing, are shown in Figure 9. The maximum amplitudes of the IPCE spectra significantly increased upon annealing, as shown in Figure 9. This feature is virtually the same as that of a homopolymer [10]. For instance, the maxima in IPCE spectra of B2 and B3 before annealing were approximately 15%, while those after annealing were increased up to approximately 55%. This improving trend of the IPCE amplitudes upon annealing is coincident with that of J_SC_. Indeed, the annealed device of H3, with the highest block ratio of the HT, showed the highest maximum IPCE of ~65% as it had the highest J_SC_ among all studied copolymer devices. In general, the highest block ratios of copolymers B3, H3, and O3 indicate a pronounced increase in the IPCE amplitude upon annealing compared to those of the copolymers with lower block ratios, B2, H2, and O2, respectively. The IPCE maxima of B3, H3, and O3 change upon annealing from ~15% to ~60%, from ~30% to ~65%, and from ~45% to ~60%, respectively. 

The maxima of the IPCE spectra are red-shifted from ~500 nm to ~530 nm, which is comparable to those of the UV-Vis spectra shown in Figure 4. However, there was a slight deviation between the UV-Vis and IPCE spectra. This is attributed to the complex reflections at various interfaces formed with a glass substrate, thin ITO transparent electrode, and light-absorbing organic active layer followed by an opaque and reflective metal electrode. These interact with the interference caused by the complex reflections of a typical device, while conventional UV-Vis spectra are usually measured with samples on a transparent substrate. This complicated optical situation in a multilayered device structure hinders the assessment of the exact photon-harvesting capacity of the active layer, and the underlying operational mechanism of an OPV device is investigated accurately. Therefore, the transfer matrix method can be used to accurately evaluate the internal absorption by an active layer and the therefore internal quantum efficiency (IQE) of charge carrier photogeneration, as described elsewhere [34]. Concludingly, there were no significant spectral changes in the IPCE spectra compared to the UV-Vis spectra, and the increase in the IPCE amplitudes upon annealing occurred over the entire spectral range. These results can be explained mainly by the improvement in charge carrier mobility and/or extraction capability rather than the improvement in the light harvesting alone [35]. Further discussion by combining these results with other observations, for example, PL quantum yield and device parameters of block copolymers is followed in the final section.

## 3. Discussion

The synthesis and analysis of triblock copolymers with various block ratios provided important insights for designing target-oriented polymer structures with well-defined molecular weights using the synthetically controllable GRIM method. The accurate ratios of the components in the synthesized copolymers were estimated using ^1^H-NMR. The ambiguity due to the overlap of the terminal methyl group signals of HT and OT could be resolved by estimating the monomer conversion efficiency using naphthalene as an internal standard. All the triblock copolymers of P3HT-*b*-P3BT-*b*-P3OT showed a block ratio that was virtually the same as its corresponding feeding ratio. Thermal analysis revealed that the block copolymers exhibited well-defined co-crystallization behavior, although they consisted of three different types of blocks. The T_c_ and T_m_ of triblock copolymers are block ratio-dependent, and the values are closer to those of the corresponding homopolymers, particularly when one block component becomes dominant to form a triblock copolymer.

UV-Vis absorption and PL emission spectra of copolymers for various sample conditions—namely, solutions, pristine films, and blended films—could be characterized before and after annealing. The spectroscopic results are clearly block ratio-dependent, although the dependency tends to be reduced upon blending, especially in conjunction with annealing treatment. However, the IPCE and PCE results seem block ratio-dependent as shown in Figure 8 and Figure 9. For a better comparison, all IPCE values at 500 nm were corrected using the corresponding reflectance values listed in Appendix A and the transmittance of ITO covered glass substrate (82.0%) to form primary IQE values. More accurate IQE values must be evaluated by considering their complex interference effect according to the Fresnel formalism in addition to the above-mentioned primary corrections. However, the obtained IQE values in this study are firmly comparable with the accurate IQE values of a similar BHJ system consisting of a narrow band gab polymer known as PTB7 and a fullerene derivative [34]. Because the devices were prepared under same conditions and the studied copolymer blend films have virtually same optical characteristics, including reflectance, especially after annealing, the internal absorption situation must be comparable. Therefore, the obtained primary IQE values must be intuitive and reliable to compare with each other despite the simple calculation process. Interestingly, the primary IQE values of polymers in Figure 10 show pronounced block ratio dependence mainly for the HT copolymer series. This block ratio dependence of the IQE is comparable to that of J_SC_ because the IQE based on the IPCE and the J_SC_ are measured under the same 0 V bias condition. This must be a clear sign that the charge carrier photogeneration facilities of all copolymers are similar and not significantly block ratio-dependent except HT series. This is also closely related to the block ratio-independent PLQY of copolymers after blending and annealing. However, the obtained PCEs of all block copolymers are block ratio-dependent as shown in Figure 8. The block ratio dependence of PCEs has remarkably analogous functional dependence with FF which means that the PCE improvement in copolymer blend systems are mainly caused by the improvement in the charge carrier mobility and/or extraction facility upon annealing.

The block copolymer with the highest portion of hexyl group, H3, exhibited the best photovoltaic performance with 4.1% of PCE among all studied devices. Actually, all types of copolymer series have shown the same block ratio-dependent PCE trend; PCE is increased when a block component becomes dominant as in the case of H3, B3, or O3. Those block copolymers with a dominant block component have significantly analogous device properties with their corresponding homopolymers although the copolymer chain contains approximately 40% of different types of monomers. For example, H3’s corresponding homopolymer P3HT was often reported to reach ~4% of its optimized PCE with analogous device parameters to those of H3. However, the best PCE of P3HT could be obtained only with MW of ~40 kDa. Conversely, the best PCE of H3, being comparable with that of P3HT, could be obtained with MW of ~15 kDa [10]. If the other 40% of types of monomer contents are considered, then the MW of a pure HT segment on an H3 copolymer chain is only ~9 kDa. It is noteworthy that the PCE of P3HT homopolymer with MW of ~15 kDa was only ~2%, although its polydispersity index and regioregularity was virtually the same as those of H3. 

The effects of the triblock copolymer devices’ upon varying the block ratio must have crucial meanings to understand the deeper insights of those conducting polymers which should be regarded as important potential candidates for flexible electronic applications. One of the reasons for this must be the possibility of synthetically obtaining structurally well-defined block copolymers with a narrow polydispersity index, as well as the possibility of combining blocks with specific properties. This would enable researchers to disentangle complex electronic behaviors caused by the impact of broad MW distribution and increasing uncertainty upon the elongation of the chain lengths of potential polymeric materials. Therefore, it can be expected that further studies related to, for example, charge carrier mobility and which make use of these well-designed block copolymers may provide promising information for research and development together.

## 4. Materials and Methods

All reactions were performed under an argon atmosphere using the Schlenk technique. 2-bromo-5-iodo-3-alkylthiophenes were prepared according to a previously established procedure with slight modifications [21]. Tetrahydrofuran (THF) and dichloromethane (CH_2_Cl_2_) were dried over Na/benzophenone and CaCl_2_. The solvents were freshly distilled before use. Other solvents, such as chlorobenzene (CB), methanol, and chloroform, were purchased from Samchun Pure Chemicals Co. (Korea) and used without further purification. Isopropymagnesium chloride solution (2.0 M in THF), [1,3-bis(diphenylphosphino)propane]dichloronickel (II) (Ni(dppp)Cl_2_), iodine, and iodobenzene diacetate (PhI(OAc)_2_) were purchased from Sigma-Aldrich. 2-bromo-3-hexylthiophene, 2-bromo-3-butylthiophene, and 2-bromo-3-octylthiophene (Puyang Huicheng Electronic Material Co., Ltd., China) were purified by flash chromatography using hexane as the eluent. Poly (3,4-ethylenedioxythiophene): poly(styrenesulfonate) (PEDOT:PSS) (Clevios P VP AI 4083) and PCBM (Nano-C Inc.) were used without further purification.


**Synthesis of the triblock copolymers.**


The molar feed ratios of each monomer are listed in Table 1. All the glass apparatuses were dried using the Schlenk line technique prior to use. For the synthesis of copolymer P1, 2-bromo-5-iodo-3-hexylthiophene (1.119 g, 3 mmol) was added to a three-neck round-bottom flask (250 mL) under an argon atmosphere. After the addition of dried THF (20 mL), the solution was stirred at 0 °C for 15 min and i-PrMgCl solution (1.5 mL) in THF (2.0 M, 3 mmol) was added via a syringe. The solution was allowed to be stirred at 0 °C for 2 h. 19.45 mg of Ni(dppp)Cl_2_ (1.2% of total molar ratio, 0.036 mmol) in THF (10 mL) was added at 0 °C. The reaction was allowed to proceed at room temperature, and polymerization was continued for 2.5 h (Flask 1). 

In another flask, 2-bromo-5-iodo-3-butylthiophene (1.035 g, 3 mmol) was reacted with a solution of *i*-PrMgCl (1.5 mL, 3 mmol) at 0 °C for 2 h. This solution was added to Flask 1 under an argon atmosphere, and the resulting mixture was continuously reacted at room temperature for 2.5 h. Similarly, 2-bromo-5-iodo-3-octylthiophene (1.203 g, 3 mmol) was used to prepare the Grignard solution. After 2 h, the solution was added to flask No.1. The resulting mixture was then stirred overnight at room temperature. The reaction was quenched with 20 mL HCl solution (5 M) and precipitated in methanol. The residual polymer was filtered and successively washed using the Soxhlet method in the order acetone, hexane, and chloroform. The chloroform fraction was concentrated under reduced pressure. The dried polymeric solid in a flask was collected by trituration with a help of adding a small portion of methanol and then the collected triturates are filtered to yield a purple product. The other copolymers with different P3AT block ratios were synthesized in the same manner to yield purple solid products. 

To determine the degree of polymerization, a similar procedure was performed in the presence of naphthalene (0.384 g, 3 mmol) [22]. After 24 h, the reaction mixture was quenched with an HCl solution (5 M) and extracted with Et_2_O. The organic layer was washed with NaHCO_3_ and NaCl and dried with anhydrous MgSO_4_. The organic layer was used to determine the degree of monomer conversion based on the amount of monomer remaining compared to naphthalene by analysis of ^1^H-NMR. 


**Fabrication of photovoltaic device.**


Photovoltaic devices with a conventional sandwich structure of indium tin oxide (ITO)/PEDOT:PSS/active layer (copolymer:PCBM)/lithium fluoride (LiF)/aluminum (Al) were fabricated to measure the PCE and spectrally resolved incident photon-to-current efficiency (IPCE). ITO-covered glass substrates (10 Ω/sq, 180 nm thickness, Samsung Corning) were ultrasonically cleaned for approximately 15 min in deionized water with <2% detergent (Mucasol, Sigma Aldrich, St. Louis, MO, USA), followed by deionized water, acetone, and isopropanol. The ITO substrates were then treated with O_2_ plasma for approximately 10 min. The PEDOT:PSS solution was filtered (0.2 µm) and deposited by spin coating onto the pre-treated ITO-covered glass at 2700 rpm for 45 s to obtain a layer with a thickness of approximately 50 nm. This layer was subsequently baked at 140 °C for 15 min under ambient conditions. Blend films for the active layer were prepared from a mixture of the synthesized copolymer: PCBM solution with a weight ratio of 1:0.7 in 1 mL of CB (2 wt %). The solution was stirred at 60 °C for 1 h and then overnight under a nitrogen atmosphere at room temperature. The active layers of the devices were spin-coated at 700 rpm for 45 s with the prepared solutions, which were filtered through a 0.45 μm syringe filter on the PEDOT:PSS layer. The thickness of the active layer was measured as 75 ± 10 nm using a surface profiler. The devices were then moved to high vacuum for drying (4 h). The top electrode was thermally evaporated with a combination of 0.3 nm LiF and 120 nm Al layers under a 10^−6^ mbar vacuum. The prepared devices have active pixel areas of 0.9 cm^2^. Detailed information about the device dimensions is presented in the Appendix A. To verify the effect of annealing time on device performance, the devices were annealed at 140 °C for 15 min. The uniformity of the obtained films was sufficient to be used as typical active layers of OPV devices. For example, conventional microscopic images of selected pristine triblock copolymer films after annealing are shown in Appendix A.


**Measurements.**


^1^H-NMR spectra were recorded on a Bruker 400 MHz FT-NMR spectrometer at room temperature using CDCl_3_ as the solvent. NMR data were reported in parts per million as chemical shifts relative to tetramethylsilane as an internal standard. The number-averaged (Mn) and weight-averaged molecular weights (Mw) of the synthesized polymers were estimated by gel-permeation chromatography (GPC) using a commercial system (Hewlett Packard, HP 1050 Series HPLC, Palo Alto, CA, USA). For the GPC measurement, HPLC-grade THF was used as the dissolving eluent and maintained at 40 °C. The flow rate was 2.0 mL min^−1^, and the injection volume was 30 μL. The system was calibrated using polystyrene standards (American Polymer Standards Co., Ltd., Mentor, OH, USA). 

Pristine thin films for various spectroscopic characterizations were prepared by spin coating the synthesized polymers onto glass substrates. The thicknesses of the thin films were measured using a surface profiler (AMBiOS, XP-200, Santa Cruz, CA, USA). UV-Vis absorption spectra (Scinco, Neosys-2000, Seoul, Korea) and PL spectra (Scinco, FluroMate FS-2, Seoul, Korea) were recorded for CB and thin films. The thermal properties were determined by differential scanning calorimetry (DSC) measurements using commercial equipment (Scinco, DSC N-650, Seoul, Korea) at heating and cooling rates of 10 °C/min under nitrogen flow. In addition, light-integrating sphere-assisted steady-state photoluminescence quantum yield (PLQY) spectra were measured using a commercial spectrometer (Hamamatsu Photonics, Quantaurus-QY, Shizuoka, Japan). 

The current density–voltage (J–V) characteristics of the OPV devices were measured using a source measurement unit (Keithley SMU 2410, Solon, OH, USA) under illumination conditions, employing a Newport 94062A solar simulator in a clean room at 25 °C. The solar simulator was calibrated with a reference cell certified by the National Renewable Energy Laboratory (NREL, USA) to maintain standard test conditions of an air mass of 1.5, spectral irradiance distribution, and radiant intensity of 1000 W/m^2^. The IPCE spectra of the OPV devices were obtained using a commercial IPCE equipment (PV Measurement Inc. QEX7, Boulder, CO, USA).

## Data Availability

Data available on request from the authors.

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
