# Peer review of "Influence of Block Ratio on Thermal, Optical, and Photovoltaic Properties of Poly(3-hexylthiophene)-*b*-poly(3-butylthiophene)-*b*-poly(3-octylthiophene)"

_molecules, 2022, doi:10.3390/molecules27238469_

Round 1
Reviewer 1 Report
The authors synthesized P3HT-b-P3BT-b-P3OT for basic material properties, and solar cell fabrication and characterization. Overall, the manuscript is well-written, and most important data are shown properly. However, some information is missing. Though the authors made the solar cells in a typical structure, the device dimensions are not shown. It is critically important in solution-processed devices because of film uniformity issues. So, I suggest that the authors add more information in the supplement or main text for a schematic diagram of device structure, actual pictures of the devices with dimensions, and optical microscope or AFM images of the surface of the active layer (copolymer:PCBM). The pictures will show the device quality used in the manuscript. Adding actual pictures of the block-copolymers powder is also highly desirable.
Reviewer 2 Report
In this manuscript, Nguyen et al. reported on synthesis of triblock copolymers via Grignard metathesis polymerization, analyzed by NMR, DSC and UV-VIS/PL studies. The presented manuscript is the result of a big and careful work. It seems to me that the article deserves to be published in its current form, despite the fact that there are a few comments:
1. It would be helpful to provide 1H qNMR acquisition parameters.
2. It is not clear, how product was collected in methanol (Synthesis of the triblock copolymers, line 455), it was used for repeated precipitation from concentrated chloroform solution or for trituration of residue?
3. Has the nickel content of the polymer been analyzed?
